# Phase Angle, Handgrip Strength, and Other Indicators of Nutritional Status in Cancer Patients Undergoing Different Nutritional Strategies: A Systematic Review and Meta-Analysis

**DOI:** 10.3390/nu15071790

**Published:** 2023-04-06

**Authors:** Desirée Victoria-Montesinos, Ana María García-Muñoz, Julia Navarro-Marroco, Carmen Lucas-Abellán, María Teresa Mercader-Ros, Ana Serrano-Martínez, Oriol Abellán-Aynés, Pablo Barcina-Pérez, Pilar Hernández-Sánchez

**Affiliations:** 1Faculty of Pharmacy and Nutrition, UCAM Universidad Católica San Antonio de Murcia, 30107 Murcia, Spain; 2Faculty of Sport, UCAM Universidad Católica de Murcia, 30107 Murcia, Spain

**Keywords:** bioimpedance, cancer therapy, radiotherapy, neoplasms, phase angle

## Abstract

Malnutrition in cancer patients is one of the most influential factors in the evolution and mortality of such patients. To reduce the incidence of malnutrition, it is necessary to establish a correct nutritional intervention. For this purpose, precise tools and indicators must be developed to determine the patient’s condition. The main objective of this systematic review and meta-analysis was to analyze the relationship between different nutritional strategies, phase angle (PA), and handgrip strength in patients with cancer, with the secondary objectives being the modification of other indicators of nutritional status, such as weight and body mass index (BMI). A systematic review of randomized clinical trials was carried out in March 2023 in the databases PubMed, Web of Science, Cochrane, and Scopus. As a risk-of-bias tool, RoB 2.0 was utilized. A total of 8 studies with a total of 606 participants were included in the analysis. A significant increase in PA was observed after the different nutritional strategies (SMD: 0.43; 95% CI: 0.10 to 0.77; *p* = 0.01; *I*^2^ = 65.63%), also detecting a significant increase in handgrip strength (SMD: 0.27, 95% CI: 0.08 to 0.47; *p* = 0.01; *I*^2^ = 30.70%). A significant increase in PA and handgrip were observed in cancer patients subjected to different nutritional strategies. These results suggest that these indicators could be used in the nutritional and functional assessment of the patients.

## 1. Introduction

Cancer is a disease characterized by the development of abnormal cells, which can appear in any area of the organism, dividing, growing, or spreading with no mechanics of control [1]. Currently, neoplastic disease remains one of the main causes of morbidity and mortality worldwide, with an estimated incidence of 18.1 million new cases by the year 2020, making it one of the diseases with the greatest impact on healthcare [2]. Therefore, its monitoring is crucial since its clinical impact is of great relevance [3]. 

It has been observed that, at the time of diagnosis, there is a high percentage of patients who present involuntary weight loss, which causes the patient to begin the treatment of the disease with an impaired nutritional status [4]. Poor nutritional status is closely related to an increase in the toxicity and complications of treatment, longer hospital admissions, failure or interruption of treatments, infections, readmissions, and reduced quality of life, and is responsible for 10–20% of mortality in oncologic patients [5]. Currently, the multidisciplinary approach is considered the best option to deal with sarcopenia and cachexia caused by cancer, recommending nutritional intervention as an essential component of the therapy, being the efficient screening and also an indispensable complement [6,7].

Consequently, several consensus documents have been published to ensure early and proper nutritional monitoring and intervention in hospitals [8,9]. 

On the other hand, different studies emphasize the importance of finding markers or diagnostic protocols capable of differentiating between the different degrees of malnutrition in patients in a specific and reliable way [10,11,12]. In recent years, the role of bioimpedance and phase angle assessment has been analyzed because it allows to evaluate the nutritional status of the patient in a simple, fast, non-invasive, and convenient way, being able to obtain prognostic values. The phase angle is derived from bioelectrical impedance analysis (BIA), which measures the opposition (impedance) to the flow of an electrical current through body tissues. The phase angle is the arctangent of the reactance and resistance values obtained during the BIA [13]. A higher PA typically indicates better cellular health and integrity, while a lower PA suggests compromised cellular function or malnutrition [14]. Research has demonstrated that PA correlates well with changes in body composition and is a reliable indicator of nutritional status [15,16]. In this way, it is useful in prevention and diagnosis, complementing the other markers and improving the specificity and diagnostic sensitivity of the tools currently used to avoid late diagnoses and states of malnutrition that may complicate the treatment and recovery process [17,18]. In fact, some investigations utilized the PA as a marker to observe the role of different treatments in cancer patients with the aim of analyzing the unfavorable modifications at the body level that a certain treatment, such as radiotherapy, chemotherapy, or surgery, may cause, thereby implementing an early nutritional intervention that helps the patients to prevent potential malnutrition [19,20,21,22,23].

In several systematic reviews performed on patients with various types of cancer, it has been observed that a low-phase angle is associated with an impaired nutritional and functional status, which may increase the morbidity and mortality of the subject suffering from the disease [24,25,26,27]. It also seems to be a good marker of nutritional status, providing additional data and contributing to more rigorous nutritional evaluations in patients with neoplasms [28]. In fact, a recent meta-analysis, which included 14 studies and 2625 participants, concluded that the phase angle could become a major prognostic factor for survival in cancer patients [24].

On the other hand, dynamometry is a widely used method to assess muscle strength and functional capacity in patients with cancer [29]. The measurement of muscle strength using dynamometry has been proposed as a potential marker of malnutrition in these patients [30]. Several studies have been conducted to investigate the association between muscle strength and malnutrition in cancer patients.

A study conducted by Barata et al. found that handgrip strength, as measured by dynamometry, was significantly correlated with nutritional status in patients with advanced lung cancer [31]. Similarly, another study by Kilgour et al. showed that handgrip strength was a significant predictor of survival in patients with advanced pancreatic cancer [32]. In addition, a recent systematic review and meta-analysis carried out by López-Bueno et al. evaluated the relationship between muscle strength and mortality in cancer patients. The review included 48 studies and concluded that muscle strength measured by dynamometry was a reliable marker of mortality in patients with cancer [33]. In summary, the functional capacity measured by dynamometry is an important marker of malnutrition in people with cancer.

Therefore, and considering that no exhaustive evaluation has been performed on the relationship between the role of nutritional intervention and phase angle or dynamometry in cancer patients, the main aim of this meta-analysis was to explore PA obtained from bioimpedance or functional capacity obtained from handgrip strength and its increase or decrease depending on the nutritional treatment used and the previous treatment, thus observing the possibility of using these methods as markers of nutritional status.

## 2. Materials and Methods

### 2.1. The Literature Search

This meta-analysis was performed following the criteria established in the Preferred Reports for Systematic Reviews and Meta-Analysis (PRISMA) items [34]. The protocol was registered in the International Prospective Registry of Systematic Reviews in progress (PROSPERO CRD42022364133).

### 2.2. Eligibility Criteria

This review was conducted to address the following question: Is there any change in phase angle and other indicators of nutritional status in cancer patients undergoing different nutritional strategies?

To establish inclusion and exclusion criteria, the PICOS method was used as follows: “P” (participants) = subjects with cancer; “I” (intervention) = Cancer patients undergoing nutritional supplementation with a nutritional intervention, “C” (comparisons) = Subjects with cancer undergoing or not a nutritional intervention; “O” (outcomes) = phase angle or handgrip strength; “S” (study design) = randomized clinical trials included.

Studies were included if the following inclusion criteria were met: (a) participants: subjects with cancer; (b) outcome: phase angle measured by bioimpedance or handgrip strength; (c) study design: randomized clinical trials were included in which there were both an experimental and control groups, and duplicates were excluded. Searching was restricted to articles published in English-language peer-reviewed journals.

Exclusion criteria were as follows: (a) subjects who did not have cancer; (b) studies in which no nutritional intervention was performed; (c) those clinical trials in which there was insufficient information reported or the impossibility of data extraction; (d) studies in which the nutritional intervention was equivalent in both groups (control and experimental), modifying only the dosage.

### 2.3. Information Sources and Search Strategy

The search was performed on 18 March 2023. Two researchers (D.V.-M. and P.B.-P.) systematically searched PubMed, LILACS, Web of Science, and Cochrane Database of Systematic Reviews databases with date limits from January 1994 to March 2023. The following search strategy was used, based on the established criteria: (a) “cancer”, “neoplasm”, “tumor”, “neoplasia”, “malignancy”; (b) “phase angle”, “handgrip”; (c) “nutritional intervention”, “oral nutrition support”, “ONS”. All search terms were adapted to the database used, using the specific filters of each of them. In addition, the references of each article were reviewed to ensure that no eligible studies were omitted.

### 2.4. Selection Process

After searching through the different databases, Zotero software (version 5.0.96.2) was used to remove duplicates. Two members of the research team (D.V.-M. and P.B.-P.) reviewed the titles and abstracts, carrying out the selection process independently and identifying the relevant articles to be screened. A third researcher (A.M.G.-M.) was responsible for solving any disagreements.

### 2.5. Data Items

For data extraction, a single author performed the first phase of the process (D.V.-M.), then a second author (P.B.-P.) confirmed the extracted information. The PICOS strategy was used to obtain the demographic information of the included population and the characteristics of each intervention [35]. The country of origin of the study, the design, the date of publication, and the results obtained by each study were also extracted as data. The included studies were classified according to the relationship between the phase angle, handgrip strength, and the nutritional intervention used. On the other hand, BMI evolution and body weight changes were also included.

### 2.6. Quality Assessment

The risk of bias was assessed using the risk-of-bias (RoB 2.0) tool proposed by the Cochrane Collaboration [36]. This tool evaluates five domains related to the published article, classifying the bias as low risk if it is unlikely to affect the results, high risk if it is likely to affect them, or some concerns when there is insufficient information for its classification. 

The publication bias was set at 0.10 and was evaluated via the Funnel plot and Egger’s test [37].

### 2.7. Synthesis Methods

To analyze the effects of the nutritional intervention on the variables studied in cancer patients, the DerSimonian and Laird method or the inverse of variance method was used, depending on whether a fixed or random effects model was utilized [38]. Forest plots were generated to graphically represent the results of each study with the corresponding 95% CI [20,39,40,41,42]. For this purpose, standardized mean differences (SMD) and 95% CI were used for all comparisons. Positive SMD values indicate an increase in phase angle. Likewise, positive values for the variable “handgrip strength”, “weight”, and “BMI” indicate a benefit in terms of functional capacity. Conversely, negative values of SMD indicate greater malnutrition and lower functional capacity. For this purpose, standardized mean differences (SMD) and 95% CI were used for all comparisons. Positive standardized mean difference values indicate an increase in phase angle. Likewise, positive values for the variable “handgrip strength”, “weight”, and “BMI” indicate a benefit in terms of functional capacity. Conversely, negative values of the standardized mean difference would indicate greater malnutrition and lower functional capacity. The combined effect size of SMD was classified as small (0–0.2), medium (0.2 to 0.5), or large (> 0.50). The *I*^2^ index was used to test the heterogeneity of the included studies, being classified as not important (<40%), moderate (40–60%), substantial (60–75%), and considerable (75–100%) [43]. To perform the statistical analyses, a fixed-effects model was used if *I*^2^ was not statistically significant (*p* > 0.05). Otherwise, a random-effects model was used. All statistical analyses were performed with Stata (Version 16.1; StataCorp., College Station, TX, USA). Statistical significance was set at *p <* 0.05.

## 3. Results

A total of 136 studies were identified in the databases as well as 3 studies were included from references of different articles (Figure 1). After elimination of duplicates and screening of the titles and abstracts of each of the articles, 18 were selected as potentially eligible for full-text reading. Ten articles were excluded, three of them for not containing valid data [44,45,46], another three because they were poster or abstract [47,48], three because of a lack of a control group [19,20,49,50], and one for including patients without cancer [51] (provided in Appendix A). Finally, eight studies [39,40,41,42,52,53,54,55] were selected for the meta-analysis.

### 3.1. Study Characteristics

The main characteristics of the nine included studies are summarized in Table 1. The articles were published between 2006 and 2023. A total of 606 participants (43.9% of women) with a mean age of 58.6 ± 11.2 years were included in the present meta-analysis. The stage of the disease was reported in all studies. Regarding BMI, the mean value was 24.0 ± 3.6 Kg/m^2^. In terms of geographical regions, five different countries were identified: Italy [39,40]; Thailand [53]; Germany [41]; Brazil [42,52,55]; Switzerland [54]. All the studies were conducted with participants from only one country. In five studies [39,40,41,42,52], a bioimpedance measurement was performed before and after the nutritional intervention, obtaining the phase angle. In six of them [39,40,41,53,54,55], functionality was analyzed via a handgrip strength test. On the other hand, the evolution of body weight was measured in seven articles [39,40,41,42,52,54,55], while BMI was only analyzed in five of them [41,42,52,53,55].

### 3.2. Nutritional Intervention

All the studies included in this meta-analysis carried out a nutritional intervention or supplementation in the experimental group. The two studies conducted by Cereda et al. provide nutritional advice in both control and experimental groups [39,40]. In one of these two studies, the patients in the experimental group, called oral nutritional supplement (ONS), received two drinks a day with a high-energy and high-protein formula enriched in omega-3 [40]. In the second study, the experimental group received two sachets daily of a whey protein (WP) formula [39]. Norman et al. also used in their clinical trial a type of nutritional intervention targeted to the experimental group based on a supplement, in this case, creatine [41]. In the clinical trial conducted by Cruz et al., an amount of 2 g of fish oil with eicosapentaenoic acid (EPA) was given to the experimental group as a possible nutritional aid in addition to a diet rich in energy and protein, which was consumed by both groups [42]. Faccio et al. [52], Uster et al. [54], and de Souza et al. [55] provided the intervention group with a high-protein and personalized diet with nutritional supplements. Finally, Thambamroong et al. used curcumin supplementation in cancer patients, comparing the results with the control [53].

### 3.3. Risk of Study Bias

The RoB 2.0 tool developed by Cochrane was used for the risk of bias. The data from the nine studies are represented in Figure 2 and Figure 3 [39,40,41,42,52,53,54,55]. Three studies reported a medium overall risk of bias due to the D5 item “selection of the reported result” [39,52,54]. In general, the included articles reported a low risk in overall risk of bias [39,40,53,55].

### 3.4. Findings from Meta-Analysis

Four meta-analyses were conducted to test the effects of different types of nutritional strategies on the variables phase angle, handgrip strength, BMI, and weight. A random-effects model was used for the phase angle variable, and a fixed-effects model for the rest of the variables.

Figure 4 shows a positive and significant correlation between phase angle and nutritional strategy (SMD: 0.43; 95% CI: 0.10 to 0.77; *p* = 0.01) (*I*^2^ = 65.62%; *p* = 0.02).

Regarding handgrip strength, the meta-analysis found significant differences between experimental group with nutritional strategy (WP; ONS; creatine; hyper-protein personalized diet or curcumin) and control group. Figure 5 shows the analysis according to handgrip strength (SMD: 0.27, 95% CI: 0.08 to 0.47; *p* = 0.01) (*I*^2^ = 30.70%; *p* = 0.21).

After performing the meta-analysis, significant differences were observed in the changes in weight (SMD: 0.25, 95% CI: 0.08 to 0.42; *p* < 0.00) (*I*^2^ = 46.85%; *p* = 0.08) (Figure 6), but no significant differences were observed in BMI (SMD: 0.10, 95% CI: −0.16 to 0.36; *p* = 0.44) (*I*^2^ = 41.30%; *p* = 0.15) (Figure 7) in the experimental group, although a trend to an increase in weight can be appreciated.

The Egger regression test showed no significant differences in phase angle, handgrip strength, and BMI (*p* > 0.1), indicating an absence of publication bias. Egger’s test revealed a statistically significant result for the weight (*p* = 0.02). However, in the evaluation of phase angle, a visual assessment using the funnel plot suggests publication bias, although the result of Egger’s test is not statistically significant (Appendix A).

## 4. Discussion 

To our knowledge, this is the first meta-analysis that has comprehensively examined the impact of different nutritional strategies on the modification of phase angle and other variables such as handgrip strength, BMI, and weight in cancer patients. The results obtained in this meta-analysis suggest that a nutritional intervention or the use of different nutritional strategies could increase the phase angle and handgrip strength in people with cancer, with a trend towards an increase in BMI. 

The pathogenesis of malnutrition and the cachexia that can result from it is complex and multifactorial. Diseases, such as cancer, and the treatments used for cancer can contribute to this type of state by reducing food intake. 

The use of effective and non-invasive nutritional assessment methods is of real importance in order to know the patient’s nutritional status as soon as possible and to try to ensure that the patient is at a normal, non-critical weight at the time of treatment. After the assessment, an appropriate nutritional intervention must be implemented. Nevertheless, occasionally, this is not enough to reach this situation, and other types of complements or strategies aimed at supplementing and helping to reduce the hypercatabolic state that the subject presents are needed [56]. This is why the use of supplements, such as omega-3, for the possible modulation of systemic inflammation [57,58] and an increase in protein intake to alleviate the increased catabolism [59] is needed.

With the aim of performing an adequate nutritional assessment, multiple methods have been used. However, it has been proven that the phase angle can be a valid index to evaluate the nutritional status in this group of patients by analyzing cellular integrity and the existing direct relationship between a high value and an adequate nutritional status [27,60]. This meta-analysis has studied the modification of this marker after different types of nutritional strategies were used in cancer patients, with a significant increase noted in intervention groups. Almeida et al. observed that the phase angle could be related to nutritional markers, such as albumin and prealbumin, and could become a more sensitive indicator of nutritional status as it is not altered in situations of physiological stress [28]. In addition, this marker appears as an important prognostic factor for survival in this type of patients, being lower values indicators of higher mortality [24].

On the other hand, the results of this meta-analysis have shown how the handgrip strength and, consequently, the functional capacity of cancer patients can be improved with different nutritional strategies. In some studies, it has been observed that phase angle seems to be positively related to handgrip strength [61], observing that people with lower handgrip strength tend to present a lower phase angle [62]. Cancer is associated with a significant decrease in lean body mass (LBM) and muscle strength [63]; therefore, phase angle and handgrip strength, used as markers of functionality as well as muscle mass, could help to predict the patient’s condition and evolution during the disease [25,64]. However, it is important to address the limitations of using hand dynamometry as a measurement tool. One notable limitation is the potential influence of the operator conducting the study on the results. The way the operator encourages the patient to exert maximum physical effort can significantly affect the measured handgrip strength [65]. This variability highlights the need for standardized protocols and training for those administering handgrip strength tests to ensure consistency and reliability of the measurements across studies. 

Finally, a meta-analysis was also performed with the variables weight and BMI to observe their evolution after the nutritional strategies were implemented. Although a tendency to increase weight was observed, neither of them was statistically increased in cancer patients. Many studies have observed that weight and BMI are not very sensitive as well as highly biased nutritional markers, as neither of them distinguishes between fat mass and lean mass and does not truly reflect the nutritional status of the patient [61,66]. It is important to mention that there was an observed improvement in weight without a corresponding change in BMI, which can be attributed to the exclusion of key studies such as Cereda [39,40] and Uster [54], where BMI analysis was not conducted. As these studies did not include these data at the end of the intervention, they were not possible to analyze. Despite the lack of weight gain in these patients, it is possible that they may have experienced improved nutritional status after the nutritional strategies followed in the clinical trials. This assertion is supported by evidence from some studies, which reported improvements in other measures of nutritional status, such as increased lean body mass or reduced fat mass [67,68]. Additionally, improvements in functional markers, such as handgrip strength, suggest that the nutritional interventions may have contributed to enhanced muscle strength and overall functionality [69,70]. Expanding upon this discussion, it is essential to consider various nutritional markers and not solely rely on weight and BMI when evaluating the effectiveness of nutritional interventions in cancer patients.

The present study has certain limitations that must be acknowledged. Firstly, there are very few clinical trials analyzing the phase angle before and after a nutritional intervention in cancer patients, and the sample size is very small. Therefore, the results obtained may be biased. On the other hand, there is great heterogeneity among the included studies (tumor stage, treatment, nutritional intervention used, among others); thus, those compared in this meta-analysis do not necessarily present a direct equivalence. For instance, Cereda et al. [40] and Cruz et al. [42] included patients in stages I–IV; Thambamroong et al. [53] and Norman et al. [41] incorporated patients in stages III and IV; de Souza et al. included patients in stages II and III [55]; Uster et al. stated that the patients in the clinical trial were in an advanced stage, without specifying it [54]; Cereda et al. involved patients in stage IV [39]; and lastly, Faccio et al. mentioned that they included different stages [52].

## 5. Conclusions

To conclude, our results show that nutritional interventions can improve phase angle and handgrip strength in cancer patients, highlighting their potential as indicators of nutritional and functional status. Increased phase angle and handgrip strength may suggest improved muscle strength and overall functionality, contributing to better patient outcomes. However, given the limitations of this study, including the small sample size and heterogeneity among the included trials, further research is necessary to establish these measures as therapeutic tools and to explore their application in clinical practice.

## Figures and Tables

**Figure 1 nutrients-15-01790-f001:**
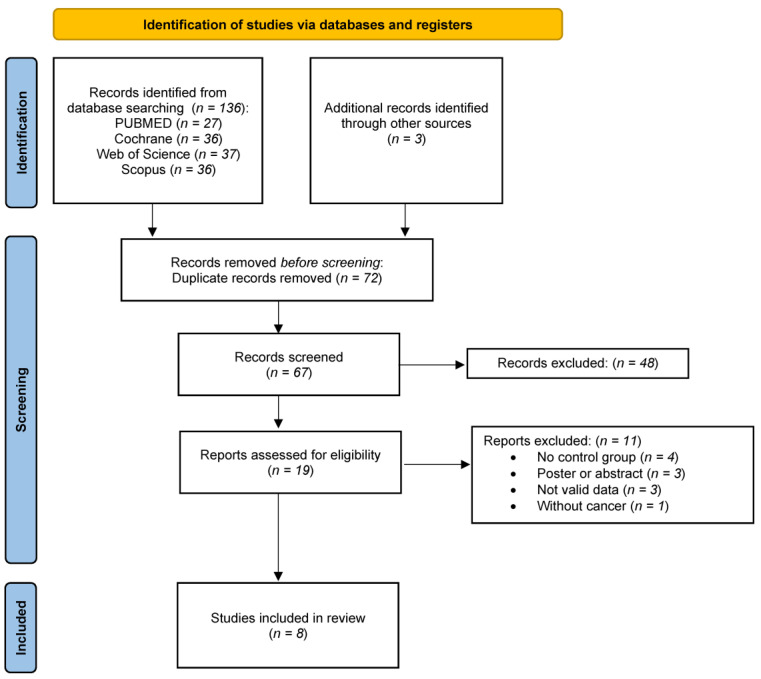
Flow diagram of the systematic review and the literature search results of the systematic review and meta-analysis.

**Figure 2 nutrients-15-01790-f002:**
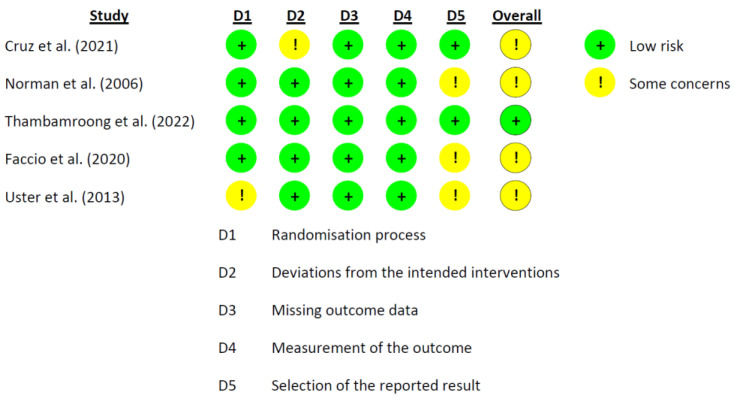
Graph of RoB 2.0 for each study according to the five domains defined by Cochrane for per protocol studies [41,42,52,53,54].

**Figure 3 nutrients-15-01790-f003:**
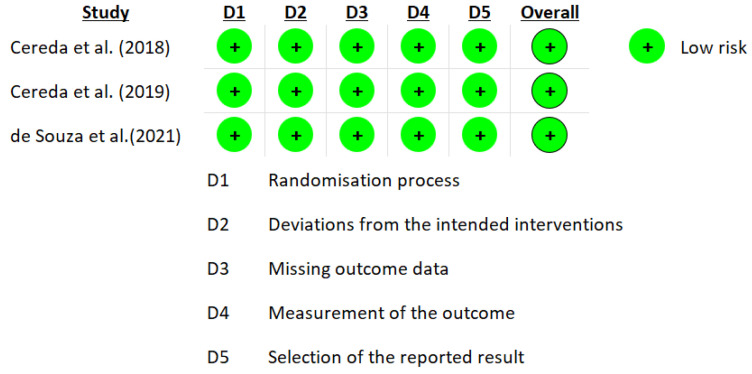
Graph of RoB 2.0 for each study according to the five domains defined by Cochrane for intention to treat studies [39,40,55].

**Figure 4 nutrients-15-01790-f004:**
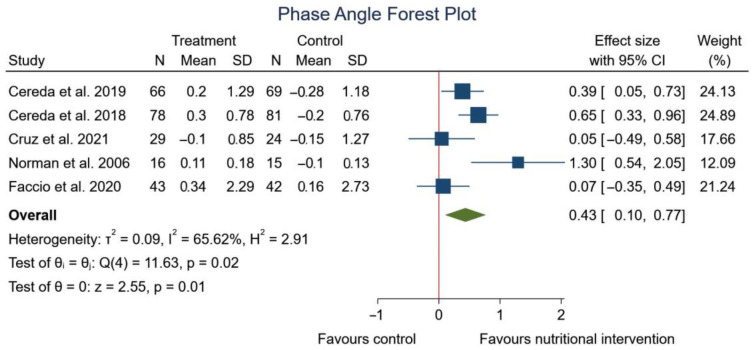
Forest plot detailing the mean difference and 95% confidence intervals (CI) for the effect of nutritional intervention on phase angle [39,40,41,42,52].

**Figure 5 nutrients-15-01790-f005:**
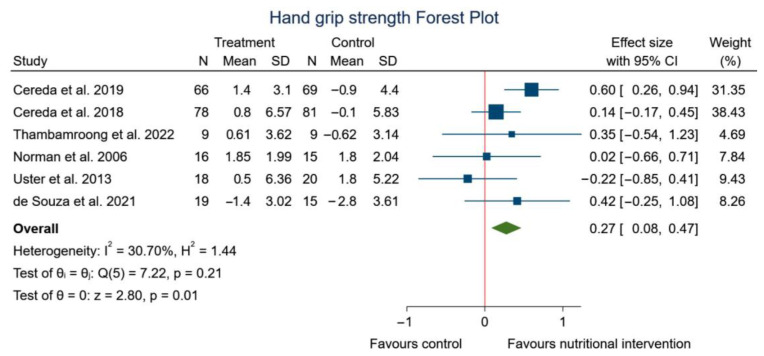
Forest plot summarizing the impact of nutritional intervention on handgrip using a fixed-effects model. Pooled summary data are presented as standardized mean differences and 95% confidence interval [39,40,41,53,54,55].

**Figure 6 nutrients-15-01790-f006:**
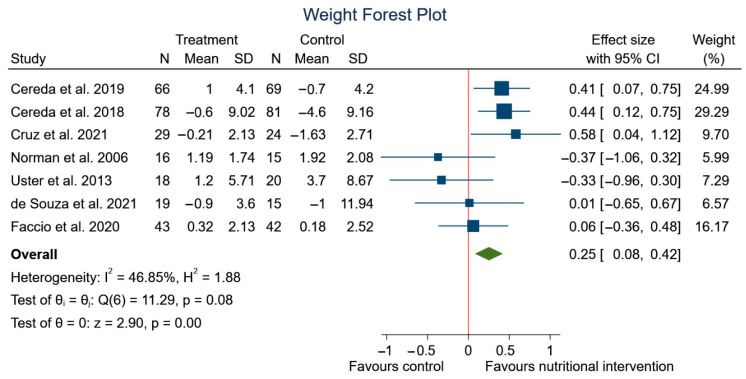
Forest plot summarizing the impact of nutritional intervention on weight using a fixed-effects model. Pooled summary data are presented as standardized mean differences and 95% confidence interval [39,40,41,42,52,54,55].

**Figure 7 nutrients-15-01790-f007:**
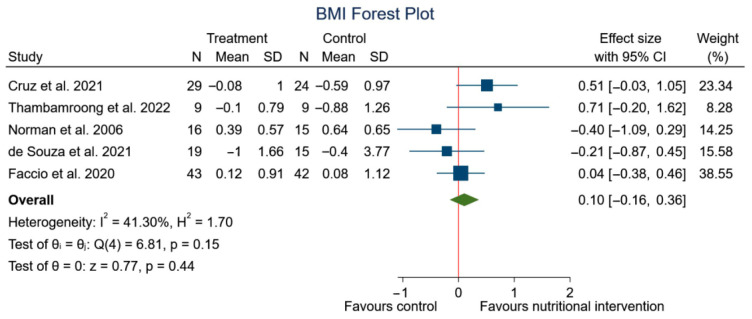
Forest plot summarizing the impact of nutritional intervention on weight using a fixed-effects model. Pooled summary data are presented as standardized mean differences and 95% confidence interval [41,42,52,53,55].

**Table 1 nutrients-15-01790-t001:** Descriptive data of the study participants (N = 606).

Reference	Year	Study Groups	Total (*n*)	Women (%)	Age (Mean)	BMI (kg/m^2^)	Cancer	Current Treatment	BIA Methods/Instrument	Nutritional Intervention	Hydraulic Hand Dynamometer
Cereda et al. [40]	2018	CS (*n =* 81) CS + ONS (*n =* 78)	159	28.3	65.1	24.2	Head and neck	Conventional (1.8-to-2 Gy/fraction) 3D conformal RT	NutriLAB, Akern/RJL	ONS (energy-dense, high-protein, omega-3-enriched oral formula)	DynEx
Cereda et al. [39]	2019	CS (*n =* 84) CS + WP (*n =* 82)	166	39.8	65.4	22.2	Lung, stomach, esophagus, pancreas, colon, blood, breast, head, and neck	Standard chemotherapy regimens	NutriLAB, Akern/RJL	Two sachets/day of cow milk WP (20 g of proteins)	DynEx
Norman et al. [41]	2006	Cr (*n* = 16) C (*n* = 15)	31	35.5	63.4	24.9	Colorectal cancer	Chemotherapy	BIA 2000 M	Creatine supplementation	Digimax electronic dynamometer
Cruz et al. [42]	2017	EPA (*n* = 29) C (*n* = 24)	53	20.8	55.5	21.6	Oral cavity	Without treatment	Biodynamics Model 450	EPA-enriched supplement from fish oil (2 g)	NR
Faccio et al. [52]	2020	C (*n* = 42) ONS (*n* = 43)	85	60	58.8	25	Colorectal, breast, lung, upper digestive tract, ovarian and other cancers	Chemo/radiotherapy	Biodynamics, 310	ONS (hyper-protein supplement, enriched with L-leucine, vitamins, and minerals	NR
Thambamroong et al. [53]	2022	C (*n* = 10) Cur (*n* = 10)	20	NR	59	NR	Head and neck cancer	Chemo/radiotherapy	InBody	Curcumin (4000 mg)	NR
Uster et al. [54]	2013	UC (*n* = 28) NT (*n* = 30)	58	20.7	65	22.8	Breast, lung, head and neck, pancreatic, colorectal, gastrointestinal, renal, prostate, and endometrium cancer. Sarcoma, lymphoma, myeloma, mesothelioma, neuroendocrine tumor, and unknown	NR	NR	Oral nutritional supplements and individual nutritional plan	Jamar
De Souza et al. [55]	2021	C (*n* = 15) NT (*n* = 19)	34	100	44.8	27.3	Breast cancer	Chemotherapy	NR	Hyper-protein personalized diet	Jamar

BMI: body mass index; C: control; Cr: creatine; CS: counseling; Cur: curcumin; EPA: eicosapentaenoic acid; NR: not reported; NT: nutritional therapy group; ONS: oral nutrition supplement; RT: radiotherapy; UC: usual care group; WP: whey protein.

## Data Availability

Not applicable.

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
