# Peer review of "Phase Angle, Handgrip Strength, and Other Indicators of Nutritional Status in Cancer Patients Undergoing Different Nutritional Strategies: A Systematic Review and Meta-Analysis"

_nutrients, 2023, doi:10.3390/nu15071790_

Round 1

Reviewer 1 Report

I found this paper interesting and will be useful to practitioners. The authors intended to analyze the relationship between phase angle and handgrip strength in patients with cancer via a meta-analysis of 8 papers found in their search. I have some suggestions for improving this manuscript.

I think the authors need to further explain phase angle for readers that are more novice in physical assessment and bioelectrical impedance analysis. How is it derived? What does it mean? How well has it been shown to correlated with changes in body composition.

Additionally, the authors need to add further discussion to the limitations of using hand dynamometry (handgrip strength), particularly how this can be significantly influenced by the operator conducting the study (e.g., how “aggressive” the person in conducting the study in gaining a maximum physical exertion effect from the patient).

It was incongruent in that weight was improved but BMI wasn’t likely due to both the Cereda studies and Uster studies not included in the BMI analysis. Although the authors acknowledged that these markers are not very sensitive in the discussion (pg 12, line 302), this section should be expanded to include omission of key studies in the BMI analysis which further supports their reluctance to use these makers.

The authors stated that it is possible that they may have had improved nutritional status (Pg 12, line 305) with a change in weight or BMI. What evidence from these studies do you have for that? Expansion of that discussion, using data from the original studies, is warranted.

Pg 12, line 310 The authors acknowledge heterogeneity among cancers and stages in terms of weight loss, it would be useful to the readers what the status of the patients used in the meta-analysis (beyond type of cancer) were discussed. Were these patients with advanced cancer (Stg III of IV), with or without a weight loss history, etc.? More information would be helpful here.

Reviewer 2 Report

Comments to the Author

In my opinion, article requires general improvement. After corrections it may be reconsidered for publication.

1. Citing literature should be at the end of the sentence: 6, 10-12, 15-19, 20-23, 31, 40-42, 15 16 45 46,47, 35-38 48-51,35 36, 36, 35, 37, 48, 49, 38 48 51, 35-38 48-51, 35 48 50, 35-37 49-51, 53 54, 24, 57, 59.

2. In the text, reference numbers should be placed in square brackets [ ], and placed before the punctuation; for example [1], [1–3] or [1,3].

3.   Revise the literature as required by the journal.

Reviewer 3 Report

A very interesting and valuable work dealing with the constantly current topic of nutrition in cancer, a clear introduction introducing further research issues. The methodology described clearly and legibly does not raise any doubts. Clear, understandable results.

The discussion is conducted critically, it is concise and practical, however, there are no conclusions, I believe that they should be prepared in the form of a few sentences that are already included in the discussion.
